# RAP: 3D Rasterization Augmented End-to-End Planning

**Lan Feng**[1,†]    **Yang Gao**[1,†]    **Éloi Zablocki**[2,‡]    **Quanyi Li**    **Wuyang Li**[1,†]    **Sichao Liu**[1,†]
**Matthieu Cord**[2,3,‡]    **Alexandre Alahi**[1,†]
[1]EPFL, Switzerland    [2]Valeo.ai, France    [3]Sorbonne Université, France
[†]`firstname.lastname@epfl.ch`  [‡]`firstname.lastname@valeo.com`

## Abstract

Imitation learning for end-to-end driving trains policies only on expert demonstrations. Once deployed in a closed loop, such policies lack recovery data: small mistakes cannot be corrected and quickly compound into failures. A promising direction is to generate alternative viewpoints and trajectories beyond the logged path. Prior work explores photorealistic digital twins via neural rendering or game engines, but these methods are prohibitively slow and costly, and thus mainly used for evaluation. In this work, we argue that photorealism is unnecessary for training end-to-end planners. What matters is semantic fidelity and scalability: driving depends on geometry and dynamics, not textures or lighting. Motivated by this, we propose *3D Rasterization*, which replaces costly rendering with lightweight rasterization of annotated primitives, enabling augmentations such as counterfactual recovery maneuvers and cross-agent view synthesis. To transfer these synthetic views effectively to real-world deployment, we introduce a *Raster-to-Real* feature-space alignment that bridges the sim-to-real gap in the feature space. Together, these components form the **Rasterization Augmented Planning (RAP)**, a scalable data augmentation pipeline for planning. RAP achieves state-of-the-art closed-loop robustness and long-tail generalization, **ranking 1st on four major benchmarks**: NAVSIM v1/v2, Waymo Open Dataset Vision-based E2E Driving, and Bench2Drive. Our results show that lightweight rasterization with feature alignment suffices to scale E2E training, offering a practical alternative to photorealistic rendering. Project page: `https://alan-lanfeng.github.io/RAP/`.

## 1 Introduction

End-to-end (E2E) autonomous driving maps raw sensory inputs directly to waypoints or control commands, offering a scalable alternative to modular pipelines (Hu et al., 2023; Jiang et al., 2023; Bartoccioni et al., 2025). Most existing methods rely on offline imitation learning (IL), where policies are trained only on expert demonstrations from large-scale logs. While effective in open-loop evaluations, such training suffers from covariate shift (Ross et al., 2011) and provides few recovery examples. Once deployed in a closed-loop, small errors cannot be corrected and quickly escalate into unrecoverable states, leaving IL-based planners brittle in practice (Chen et al., 2024).

A natural solution is to augment training with synthetic scenarios. Recent work has explored photorealistic digital twins built with 3D neural rendering, such as NeRFs (Mildenhall et al., 2020) or Gaussian Splatting (Kerbl et al., 2023), as well as engine-based simulators (Dosovitskiy et al., 2017; Li et al., 2023b). These pipelines enable counterfactual augmentation and closed-loop training by synthesizing inputs beyond the logged trajectory. By producing synthetic views that are visually indistinguishable from real images (Figure 1, left), these techniques achieve superior fidelity compared to simulator-based reconstructions (Li et al., 2023a), which also enable closed-loop rollout but fail to match visual appearance for the E2E model inference. However, despite their visual fidelity, they remain prohibitively slow and costly, making large-scale training impractical. In practice, they are mostly restricted to policy evaluation (Ljungbergh et al., 2024; Cao et al., 2025; Zhou et al., 2024; Jiang et al., 2025).

Figure 1: **Comparison of rendering paradigms for end-to-end driving. Neural or engine-based methods** (**left**) aim to minimize the sim-to-real gap in *pixel space*, but incur high computational cost. In contrast, **our approach** (**right**) leverages *3D rasterization*, which is scalable and fully controllable, and aligns rasterized inputs with real images in *feature space*.

In this work, we take a different stance: robust E2E driving training does not require photorealistic rendering, but rather semantic accuracy and scalability. Driving decisions fundamentally depend on geometry, semantics, and multi-agent dynamics rather than visual details like textures or lighting. Moreover, humans readily transfer driving skills between video games and the real world, suggesting that aligning latent task-relevant features is more important than pixel-level appearance. These observations motivate us to favor lightweight rendering, combined with feature-space alignment, as a more scalable and transferable alternative to costly photorealistic approaches.

To this end, we introduce a *3D rasterization pipeline* that reconstructs driving scenes by projecting annotated primitives (such as lane polylines and agent cuboids) into perspective views (Figure 1, right). Unlike neural/engine rendering, rasterization is training-free, fast, and highly controllable, while still capturing the semantic and dynamic cues necessary for driving. This design enables us to go beyond the fixed dataset by generating non-trivial data augmentations, including: (1) *Recovery-oriented perturbations*, where ego trajectories are perturbed to simulate recovery maneuvers, directly targeting IL brittleness; (2) *Cross-agent view synthesis*, where scenes are re-rendered from other agents' viewpoints, expanding both scale and interaction diversity. Moreover, in contrast to neural rendering, which seeks to minimize the gap in pixel space, we propose a *Raster-to-Real (R2R) alignment* module that minimizes the gap in feature space, where semantic and geometric structures are more compact and easier to align. Together, these components form **Rasterization Augmented Planning (RAP)**, a scalable data augmentation framework for robust E2E planning.

We equip RAP with multiple planners, showing that its benefits hold across diverse model designs. Extensive experiments show that RAP achieves strong **closed-loop robustness and long-tail generalization**, ranking *1st place* on the following **4 major E2E planning benchmarks**: NAVSIM v1/v2 (Dauner et al., 2024; Cao et al., 2025), Waymo Open Dataset Vision-based E2E Driving (WOD-E2E) (Xu et al., 2025), and Bench2Drive (Jia et al., 2024). We find that efficient rasterization with feature-space alignment provides the scalability and robustness needed for end-to-end planning, without requiring photorealistic rendering.

Our work makes the following contributions:

- A scalable *3D rasterization pipeline* that reconstructs driving scenes from annotations by projecting geometric primitives into camera views.
- A *Raster-to-Real (R2R) alignment* module that bridges rasterized and real inputs in feature space, through a combination of distillation and adversarial adaptation.
- The *3D Rasterization Augmented end-to-end Planning (RAP)* framework, which augments imitation learning with counterfactual scene generation and cross-agent view synthesis, achieving state-of-the-art closed-loop robustness and long-tail generalization on multiple benchmarks.

## 2 RELATED WORK

**End-to-end planning.** E2E motion planners map sensory observations directly to future trajectories or controls. Reinforcement learning approaches (Chekroun et al., 2023; Toromanoff et al., 2020; Liang et al., 2018) have been explored in simulators but remain bottlenecked by sample inefficiency and the need for scalable environments. Imitation learning (IL) from real-world expert driving logs (Pomerleau, 1988; Buhet et al., 2020; Hu et al., 2022; Chitta et al., 2023; Hu et al.,

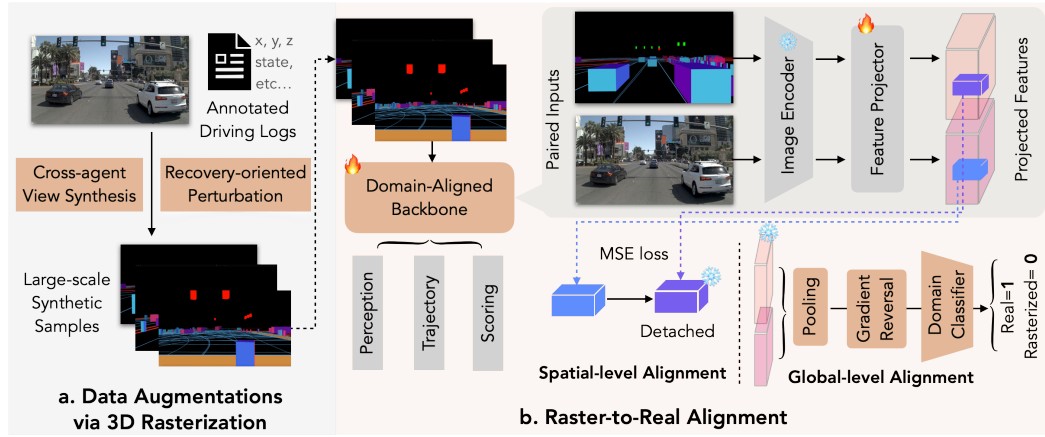

Figure 2: **Overview of the proposed RAP**. (a) **Data Augmentations via 3D Rasterization**: annotated driving logs are converted into large-scale synthetic samples through *cross-agent view synthesis* and *recovery-oriented perturbation*. (b) **Raster-to-Real Alignment**: paired real and rasterized inputs are processed by a frozen image encoder and a learnable feature projector. Spatial-level alignment uses MSE loss against detached raster features, while global-level alignment employs a gradient reversal layer and domain classifier to enforce domain confusion.

2023) is more widely adopted and has achieved strong performance in open-loop evaluations (Jiang et al., 2023; Guo et al., 2025; Liao et al., 2025). However, IL-trained policies suffer from covariate shift: they lack examples of recovery from mistakes and fail to generalize to rare long-tail events (Chen et al., 2024). To mitigate this, prior work has explored adversarial scene generation with challenging traffic behaviors (Bansal et al., 2019; Hanselmann et al., 2022; Rempe et al., 2022; Yin et al., 2024), but these efforts are restricted to a bird's-eye view and have only been validated in mid-to-end planning. In contrast, our work tackles the end-to-end setting from camera inputs, using scalable 3D rasterization to generate diverse recovery and counterfactual scenarios.

**Rendering for Driving.** Classical simulators such as CARLA (Dosovitskiy et al., 2017), LGSVL (Rong et al., 2020), and MetaDrive (Li et al., 2023b) rely on physically based rendering (PBR) with handcrafted 3D assets, but creating diverse, realistic worlds is costly, and modeling traffic behavior remains challenging. MetaDrive (Li et al., 2023b) instead synthesizes digital scenes from real-world logs, while VISTA (Amini et al., 2020; 2022) reprojects real images to nearby viewpoints, though only for small ego deviations. Neural rendering approaches such as NeRF (Mildenhall et al., 2020) and 3D Gaussian Splatting (3DGS) (Kerbl et al., 2023) reconstruct logs with high fidelity and support counterfactual replay, but suffer from poor scalability, costly optimization, and visual artifacts when views deviate significantly. Voxel- and occupancy-based reconstructions (Huang et al., 2023; Jiang et al., 2024; Li et al., 2025; Chambon et al., 2025) trade off fidelity for efficiency but require dense labeling (Huang et al., 2024). Attempts at training planners with these renderings, e.g., RAD (Gao et al., 2025), remain small-scale and photorealism-focused. Closed-loop or pseudo-closed-loop evaluations have been explored in NeuroNCAP (Ljungbergh et al., 2024), HUGSIM (Zhou et al., 2024), RealEngine (Jiang et al., 2025), and NAVSIM v2 (Cao et al., 2025), but all rely on expensive photorealistic pipelines. In contrast, we advocate lightweight rasterization, which preserves semantic and geometric accuracy, avoids pixel-level artifacts, and enables scalable and non-trivial data augmentation.

## 3 3D RASTERIZATION AUGMENTED PLANNING

RAP builds on real-world driving logs by generating additional training data that goes beyond expert demonstrations. Instead of aiming for photorealistic images, our goal is to capture the geometry, semantics, and dynamics that matter for driving. To this end, we design a lightweight rasterization pipeline that reconstructs controllable views of traffic scenes (Section 3.1), enabling fast and

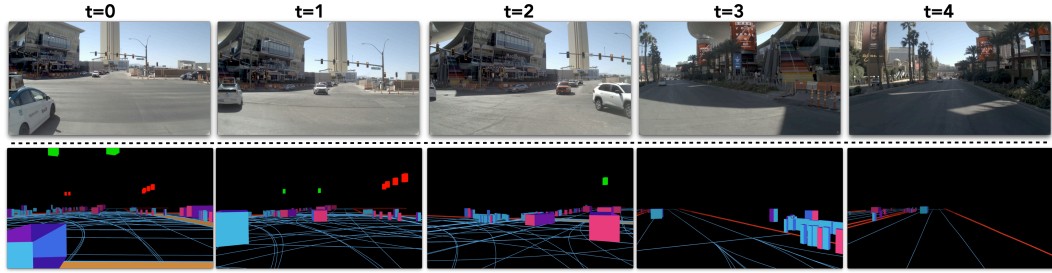

Figure 3: **Real vs. rasterized views across 5 seconds.** Top row: real front-camera inputs. Bottom row: corresponding rasterized views produced by our pipeline. Rasterization retains scene geometry and agent dynamics while discarding unnecessary appearance details.

large-scale data augmentation (Section 3.2), and we further introduce a feature-alignment module to ensure the network can effectively transfer from rasterized inputs to real images (Section 3.3).

## 3.1 3D RASTERIZATION

Our design prioritizes rendering speed and scalability, enabling the generation of large-scale augmented views. Instead of photorealism, we focus on preserving the geometric and semantic cues most relevant for driving (such as agent positions, orientations, and interactions with the map) while discarding textures and lighting details that do not affect planning.

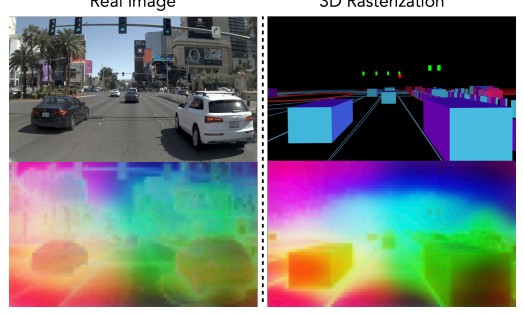

Figure 4: **PCA visualization of frozen DI-NOv3 features.** It shows rasterized and real inputs share similar structures, supporting abstraction as a perceptually valid substitute.

**Scene representation.** At each log frame we reconstruct the scene from annotations. Static map elements (e.g., road surfaces, crosswalks) are represented as polylines $\mathcal{M} = \{\mathbf{P}_k\}$ in world coordinates, where each $\mathbf{P}_k \in \mathbb{R}^{n_k \times 3}$ denotes a polyline with $n_k$ vertices in the $(x, y, z)$ plane.

Traffic-relevant objects (vehicles, bicycles, pedestrians, traffic cones, barriers, construction signs, and generic objects) are approximated by oriented cuboids

$$\mathcal{B}_i = (l_i, w_i, h_i, \mathbf{T}_i), \quad \mathbf{C}_i = \mathbf{T}_i \begin{bmatrix} \pm l_i/2 & \pm w_i/2 & 0, h_i \end{bmatrix}^\top,$$

where $l_i, w_i, h_i$ are the length, width, and height of actor $i$, $\mathbf{T}_i \in SE(3)$ is its rigid-body pose in world coordinates, and $\mathbf{C}_i \in \mathbb{R}^{8 \times 3}$ gives the eight 3D corner points of the cuboid. Traffic lights are modeled as upright cuboids with fixed dimensions, color-coded according to their state (red, yellow, green).

**World-to-image projection.** We model the ego camera as a standard pinhole camera with intrinsics $K \in \mathbb{R}^{3 \times 3}$ and extrinsics $\mathbf{T}_{w \to c} \in SE(3)$. Any 3D point $\mathbf{p}_w \in \mathbb{R}^3$ is lifted to homogeneous coordinates $\tilde{\mathbf{p}}_w = [\mathbf{p}_w^\top, 1]^\top \in \mathbb{R}^4$, and projected to the image plane as

$$\mathbf{u}_{uv} = \pi(\mathbf{p}_w) = K \mathbf{T}_{w \to c} \tilde{\mathbf{p}}_w, \tag{1}$$

After perspective division, the pixel coordinates are

$$(u, v) = \left( \frac{u_x}{u_z}, \frac{u_y}{u_z} \right),$$

with depth $u_z$. Points with $u_z < z_{\text{near}}$ are discarded.

**Rasterization.** All primitives are rasterized into an RGB canvas $\mathbf{I} \in \mathbb{R}^{H \times W \times 3}$ using depth-aware compositing, where each fragment stores depth $d$ with a fading weight $\alpha = \max(0, 1 - d/d_{\max})$ blended through a single buffer to resolve occlusions; primitives crossing the view boundary are clipped with the Sutherland–Hodgman (Sutherland & Hodgman, 1974) operator to ensure stable visibility. Focusing on geometric and semantic fidelity while discarding photorealistic details, our rasterizer delivers spatial cues essential for planning at a fraction of the cost, enabling large-scale counterfactual generation. As shown in Figure 4, features extracted by a frozen DINOv3 encoder and visualized via PCA remain qualitatively consistent between rasterized and real images, indicating that abstraction preserves perceptual cues crucial for downstream learning. Figure 3 shows a qualitative comparison between real and rasterized views across a 5 s horizon.

### 3.2 Data Augmentations via 3D Rasterization

Beyond rendering ego-centric views, our rasterization pipeline naturally supports diverse non-trivial augmentation strategies, allowing us to expand the training corpus and expose the planner to richer and rarer distributions of driving scenarios. We focus on two complementary techniques (Figure 2, left): *Recovery-oriented perturbations* to simulate recovery from off-distribution states, and *cross-agent view synthesis* to diversify viewpoints and interactions and thereby extract more value from an annotated log.

**Recovery-oriented perturbations.** We enrich the dataset by perturbing the logged ego trajectory. Given a ground-truth trajectory $\tau^*(t)$, we apply controlled lateral and longitudinal offsets together with stochastic noise:

$$\tilde{\tau}(t) = \tau^*(t) + \delta_{\text{lat}}(t) + \delta_{\text{long}}(t) + \epsilon_t,$$

where $\delta_{\text{lat}}, \delta_{\text{long}}$ are perturbations sampled from predefined ranges and $\epsilon_t$ denotes Gaussian noise. The perturbed trajectory is then re-rendered with 3D rasterization, creating counterfactual scenes in which the ego vehicle drifts away from the expert path. These samples encourage the planner to recover from distribution shifts, improving robustness in closed-loop evaluation.

**Cross-agent view synthesis.** Each nuPlan traffic scenario contains trajectories for $n$ agents (including the ego). Instead of rendering only from the ego perspective, we replace the ego trajectory with that of another agent while keeping the original camera parameters fixed. This produces realistic views from other agents without requiring new sensors.

Together, these augmentations scale beyond the limitations of logged data, yielding over $500k$ rasterized training samples that cover diverse viewpoints, richer interactions, and rare recovery scenarios.

### 3.3 Raster-to-Real Alignment

Synthetic rasters already produce features similar to real images (Figure 4), suggesting they can strengthen learning when combined. To make this benefit reliable, we introduce **Raster-to-Real (R2R) alignment** (Figure 2, right), enforcing feature consistency between rasterized and real inputs at both spatial and global levels.

**Spatial-level alignment.** For each real sample $x^r$ with a paired rasterized rendering $x^s$, we use a visual encoder $\phi(\cdot)$ to extract projected features:

$$F^r = \phi(x^r), \quad F^s = \phi(x^s), \quad F^r, F^s \in \mathbb{R}^{N \times d'},$$

where $N$ denotes the number of spatial locations (patch tokens for ViT or feature-map positions for CNNs), and $d'$ is the projected feature dimension. We freeze the raster features $F^s$ and update the real features $F^r$ by minimizing a mean-squared alignment loss:

$$\mathcal{L}_{\text{spatial}} = \frac{1}{N} \sum_{j=1}^{N} \|F_j^r - F_j^s\|_2^2. \tag{2}$$

Since raster features come from high-quality annotations and omit distracting details, they serve as a strong proxy for perception. With their abundance compared to real images, aligning real features to them offers cleaner and denser supervision.

Table 1: **NAVSIM v1 benchmark** (`navtest`). Bold/underlined indicates the best/second-best.

| Method | Input | NC ↑ | DAC ↑ | TTC ↑ | Comf. ↑ | EP ↑ | PDMS ↑ |
|---|---|---|---|---|---|---|---|
| PDM-Closed (Rule-based) (Dauner et al., 2023) | Perception GT | 94.6 | 99.8 | 86.9 | 99.9 | 89.9 | 89.1 |
| *Human* | *—* | *100* | *100* | *100* | *99.9* | *87.5* | *94.8* |
| Hydra-MDP-$\mathcal{V}_{8192}$-W-EP (Li et al., 2024) | Cam & Lidar | 98.3 | 96.0 | 94.6 | 100 | 78.7 | 86.5 |
| DiffusionDrive (Liao et al., 2025) | Cam & Lidar | 98.2 | 96.2 | 94.7 | 100 | 82.2 | 88.1 |
| DiffusionDrive-Camera Only (Reproduced) | Camera | 97.9 | 94.6 | 93.6 | 100 | 80.7 | 86.0 |
| PARA-Drive (Weng et al., 2024) | Camera | 97.9 | 92.4 | 93.0 | 99.8 | 79.3 | 84.0 |
| AutoVLA (Zhou et al., 2025) | Camera | 98.4 | 95.6 | **98.0** | 99.94 | 81.9 | 89.1 |
| iPad (Guo et al., 2025) | Camera | 98.6 | 98.3 | 94.9 | 100 | 88.0 | 91.7 |
| Centaur (Sima et al., 2025) | Camera | **99.2** | 98.7 | **98.0** | 99.97 | 86.0 | 92.1 |
| **RAP-DiffusionDrive-Camera Only** | Camera | 98.5 | 98.5 | 95.4 | 100 | 83 | 89.2 |
| **RAP-iPad** | Camera | 98.2 | 98.6 | 94.6 | 100 | 90.1 | 92.5 |
| **RAP-DINO** (Ours) | Camera | 99.1 | **98.9** | 96.7 | **100** | **90.3** | **93.8** |

Table 2: **Public leaderboard for the NAVSIM v2 benchmark** (navhard). Bold indicates the best result.

| Method | Stage | NC ↑ | DAC ↑ | DDC ↑ | TLC ↑ | EP ↑ | TTC ↑ | LK ↑ | HC ↑ | EC ↑ | EPDMS ↑ |
|---|---|---|---|---|---|---|---|---|---|---|---|
| LTF (Chitta et al., 2023) | 1 | 96.22 | 79.56 | **99.11** | 99.56 | **84.12** | 95.11 | 94.22 | **97.56** | **79.11** | 23.12 |
| | 2 | 77.76 | 70.21 | 84.29 | **98.07** | 85.14 | 75.67 | 45.41 | 95.75 | 75.98 | 23.12 |
| **RAP-DINO** (Ours) | 1 | **97.11** | **94.44** | 98.78 | **99.78** | 83.86 | **96.89** | **94.67** | 96.44 | 66.22 | **36.93** |
| | 2 | **83.20** | **83.88** | **87.39** | 98.02 | **86.87** | **80.39** | **52.27** | 95.18 | 52.41 | **36.93** |

**Global alignment.** Global alignment complements spatial-level supervision by keeping the overall feature distributions consistent. For instance, rasterized views may contain pure black backgrounds absent in real data; aligning globally mitigates such biases and improves generalization. Besides, since rasterized samples vastly outnumber real–raster pairs, unsupervised domain adaptation (Ganin & Lempitsky, 2015) provides an effective way to exploit the full synthetic corpus by enforcing global consistency even without paired supervision.

For each sample, we compute a global representation $g \in \mathbb{R}^{d'}$ by average pooling its feature map $F \in \mathbb{R}^{N \times d'}$. We then train a domain classifier $D$ to predict whether $g$ comes from real or rasterized data. Following Ganin & Lempitsky (2015), a gradient reversal layer is inserted before $D$, such that the encoder is optimized to maximize domain confusion while $D$ is optimized to minimize classification error with $y \in \{0, 1\}$ the domain label:

$$\mathcal{L}_{\text{global}} = - \mathbb{E}_{(g,y)} \big[ y \log D(g) + (1 - y) \log(1 - D(g)) \big]. \tag{3}$$

**Overall objective.** The final training objective combines task supervision with both levels of R2R alignment with $\mathcal{L}_{\text{task}}$ the total planning loss (Guo et al., 2025): $\mathcal{L} = \mathcal{L}_{\text{task}} + \lambda_s \mathcal{L}_{\text{spatial}} + \lambda_g \mathcal{L}_{\text{global}}$, where $\lambda_s, \lambda_g$ control the strength of spatial-level and global alignment.

## 4 EXPERIMENTS

We build our rasterized data from OpenScene, a compact distribution of the nuPlan dataset (Caesar et al., 2021) that contains over **1,200 hours** of annotated driving logs. Among these, about **120 hours** provide ego-centric real camera sensors. We rasterize both the ego vehicle and other agents' trajectories across all 1,200 hours of logs. To diversify recovery behaviors, we additionally perturb ego trajectories in a randomly selected **10%** subset of ego logs.

**Dataset Curation.** We extract 7-second clips, using the first 2 seconds as input and the following 5 seconds as output. For ego trajectories, we follow the NAVSIM (Dauner et al., 2024)'s Planning-aware Driving Metric Score (PDMS) filtering strategy, removing trivial cases where the constant-velocity baseline already has high scores and human demonstrations have low scores. For other vehicles, since nuPlan lacks route annotations and PDMS cannot be computed, we instead filter clips by ADE of the constant-velocity baseline (ADE > 0.5) and ensure validity. After filtering, our curated dataset consists of $85k$ samples with paired real and rasterized sensors, $8.5k$ rasterized

Table 3: **Top-6 entries on the public leaderboard** for the WOD-E2E Driving Challenge (up to September 2025).

| Method | ADE@5s ↓ | ADE@3s ↓ | RFS (Spotlight) ↑ | RFS (Overall) ↑ |
|---|---|---|---|---|
| DiffusionLTF | 2.89 | 1.36 | 6.41 | 7.72 |
| HMVLM | 3.07 | 1.33 | 6.73 | 7.85 |
| UniPlan | 2.84 | 1.27 | 6.92 | 7.78 |
| ViT-Adapter-GRU | 2.89 | 1.44 | 6.67 | 7.99 |
| Poutine (Rowe et al., 2025) | 2.74 | 1.21 | 6.89 | 7.99 |
| **RAP-DINO** (ours) | **2.65** | **1.17** | **7.20** | **8.04** |

samples with perturbations, $272k/200k$ rasterized samples from ego trajectories and other vehicles' trajectories.

**Models.** We instantiate our RAP framework with a model we call **RAP-DINO**, which combines a frozen **DINOv3-H + backbone** (Siméoni et al., 2025) with a learnable MLP projector and an iterative deformable attention decoder adapted from (Guo et al., 2025). The full model size is $\sim$888M parameters. For ablation and efficient closed-loop inference on Bench2Drive, we also introduce a lightweight **RAP-ResNet** variant with only $\sim$29M parameters. In addition, to demonstrate the model-agnostic nature of RAP, we apply our framework to existing architectures, yielding **RAP-iPad** (Guo et al., 2025) and **RAP-DiffusionDrive** (Liao et al., 2025).

**Training.** We train on the curated dataset with two supervision heads: a **multi-modal trajectory head** supervised by future trajectories, and a **trajectory scoring head** trained with PDMS scores. Training uses 4 H100 GPUs and takes about 80 hours. The trained model is directly evaluated on NAVSIM v1/v2 benchmarks. For Waymo, we fine-tune on the train/val split using the official RFS scorer. For Bench2Drive, we follow the official `bench2drive-base` dataset and augment it with our **85k real–raster pairs** and **8.5k perturbed samples**. In this case, we use a smaller ResNet34 (He et al., 2016) image backbone (**RAP-ResNet**) for faster closed-loop inference. See Section A.4 for more details.

### 4.1 LEADERBOARD EVALUATION

**NAVSIM v1/v2.** The NAVSIM benchmarks (Dauner et al., 2024; Cao et al., 2025) evaluate a planner's ability to generate a safe and efficient 4-second future trajectory (sampled at 2Hz) based on 2 seconds of historical ego states and multi-view camera inputs. The benchmark is derived from the NuPlan dataset, specifically curating challenging scenarios with intention changes while filtering out trivial straight-driving segments. NAVSIM v1 employs the PDMS, a composite score aggregating sub-metrics such as *No at-fault Collision (NC)*, *Drivable Area Compliance (DAC)*, *Time-to-Collision (TTC)*, *Ego Progress (EP)*, and *Comfort (Comf.)*. To better assess closed-loop robustness, NAVSIM v2 introduces the Two-stage *Extended PDMS (EPDMS)*, which incorporates additional criteria: *Traffic Light Compliance (TLC)*, *Driving Direction Compliance (DDC)*, *Lane Keeping (LK)*, and *Extended Comfort (EC)*. In its second stage, NAVSIM v2 further employs *3D Gaussian Splatting (3DGS)* to synthesize counterfactual camera views after policy deviations, thereby simulating closed-loop evaluation from logged data. We report results on the `navhard` split.

Table 1 shows the NAVSIM v1 benchmark, where our RAP-DINO achieves the highest overall PDMS score of **93.8**, surpassing all prior camera-based methods. Beyond our strongest model, applying RAP to existing architectures also brings consistent gains: RAP-DiffusionDrive improves PDMS by **+3.2** over the original DiffusionDrive, while RAP-iPad yields a **+0.7** improvement over iPad. These results confirm that RAP is both effective in its own right and broadly beneficial when applied to other state-of-the-art planners. RAP-DINO also sets a new state-of-the-art on NAVSIM v2 (Table 2), achieving an overall **EPDMS** of 36.9, substantially higher than LTF, while maintaining strong performance across both Stage 1 and the more challenging Stage 2 counterfactual evaluation.

**WOD Vision-based E2E Driving.** Given 4 s of multi-camera inputs plus ego history and route, the task is to predict a 5 s future trajectory, evaluated and ranked by the *Rater Feedback Score*. The dataset includes 4021 segments (2037 train, 479 val, and the rest test) curated for long-tail events

Table 4: **Closed-loop Results on Bench2Drive Benchmark** (Jia et al., 2024).

| Method | Efficiency ↑ | Comfortness ↑ | Success Rate (%) ↑ | Driving Score ↑ |
|---|---|---|---|---|
| AD-MLP (Zhai et al., 2023) | 48.45 | 22.63 | 0.00 | 18.05 |
| UniAD-Tiny (Hu et al., 2023) | 123.92 | **47.04** | 13.18 | 40.73 |
| UniAD-Base (Hu et al., 2023) | 129.21 | 43.58 | 16.36 | 45.81 |
| VAD (Jiang et al., 2023) | 157.94 | 46.01 | 15.00 | 42.35 |
| DriveTransformer (Jia et al., 2025) | 100.64 | 20.78 | 35.01 | 63.46 |
| iPad (Guo et al., 2025) | 161.31 | 28.21 | 35.91 | 65.02 |
| **RAP-ResNet** (Ours) | **165.47** | 23.63 | **37.27** | **66.42** |

Table 5: **Ablation study on 3D rasterization design choices** (Section 4.2).

| ID | Face Rendering | Depth Decay | Background | MinADE ↓ |
|---|---|---|---|---|
| A | Colored | Yes | Black | **0.91** |
| B | Transparent | Yes | Black | 0.98 |
| C | Colored | No | Black | 1.05 |
| D | Colored | Yes | Natural | 1.33 |

Table 6: **Ablation on recovery-oriented perturbations.** Evaluation is conducted on NAVSIM v1 and v2 (Section 4.2).

| Training Set | NAVSIM v1 ↑ PDMS | NAVSIM v2 ↑ two-stage EPDMS |
|---|---|---|
| W/o Perturbed | **92.5** | 32.5 |
| Perturbed | **92.5** | **36.9** |

such as construction detours, pedestrian accidents, and unexpected freeway obstacles. These cases occur with a frequency below 0.003% in daily driving, underscoring the dataset's focus on rare but safety-critical scenarios.

Table 3 shows the results. Our RAP-DINO achieves the best overall results, with an *RFS (Overall)* of 8.04 and the lowest *ADE@5s* (2.65) and *ADE@3s* (1.17). Notably, RAP surpasses *Poutine* (Rowe et al., 2025), a strong 3B-scale vision–language model, while also attaining the highest *RFS (Spotlight)* of 7.20, significantly outperforming all competing entries. These results highlight RAP's strong generalization and robustness in long-tail scenarios.

**Bench2Drive (Jia et al., 2024).** We use the CARLA (Dosovitskiy et al., 2017) simulator, employing the Bench2Drive benchmarks for closed-loop performance evaluation of RAP. Bench2Drive provides 1,000 clips from the expert model Think2Drive, with 950 for training and 50 for open-loop evaluation. Closed-loop evaluation covers 220 routes across CARLA towns, each containing a safety-critical event, and reports four metrics: *Success Rate*, *Driving Score*, *Efficiency*, and *Comfortness*. As shown in Table 4, our RAP-ResNet achieves the best overall performance, attaining the highest *Efficiency* (165.47) and *Driving Score* (66.42), as well as the highest *Success Rate* (37.27%). These results highlight that RAP significantly improves route completion and robustness in closed-loop evaluation, establishing a new state-of-the-art on Bench2Drive.

## 4.2 ABLATION STUDY

Unless otherwise specified, all ablations are conducted on the `navtrain` subset of NAVSIM. The training set consists of **85k** paired samples with both real and rasterized views, along with an additional **100k** raster-only samples generated from ego trajectories. Evaluation is performed on the validation split (**12k** samples) using Minimum Average Displacement Error (*MinADE*) as the metric. We use RAP-ResNet for efficient training and accelerated experimentation.

**3D Rasterization Design Choices.** We ablate three factors in the rasterization pipeline: whether the object cuboids use solid colors or semi-transparent faces, whether to apply a depth-based intensity decay, and whether the background is pure black or a natural sky–ground split. See appendix (Section A.2) for visualizations.

Results in Table 5 show that solid-colored faces, depth decay, and a black background together achieve the best ADE. Removing any of these components leads to noticeable degradation, confirming that semantic cues, depth-aware rendering, and minimal background distractions are all important for effective learning.

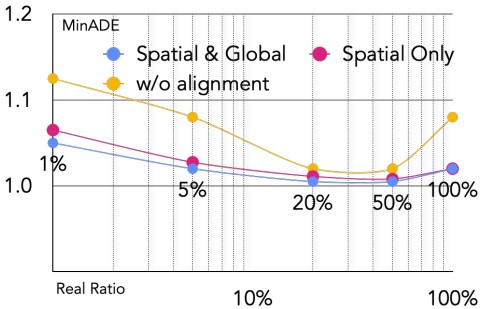

Figure 5: **Ablation on R2R alignment** (Section 4.2), showing that both spatial and global alignment improve performance.

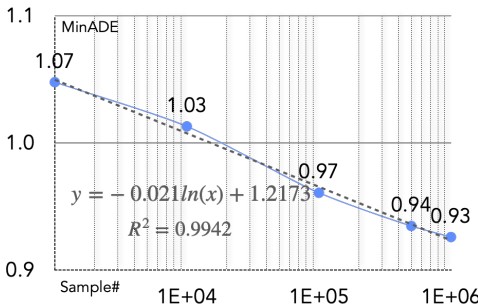

Figure 6: **Scaling curve for cross-agent view synthesis** (Section 4.2), showing consistent gains as more synthetic samples are added.

**Ablation on Recovery-oriented Perturbations.** We further examine the effect of recovery-oriented perturbation, which exposes the planner to counterfactual recoveries during training. Two training sets are compared: the `navtrain` split with **85k** real samples, and the same split augmented with **8.5k** perturbed ego trajectories. Since this modification is designed to improve closed-loop robustness, we evaluate on NAVSIM v1 and v2 using the PDM score. Table 6 shows that perturbation has no effect on v1 (both 92.5), but significantly improves v2 from 32.5 to 36.9. This is expected, as NAVSIM v2 adopts a two-stage evaluation protocol that better reflects closed-loop performance.

**Ablation on R2R Alignment.** We evaluate R2R alignment on the `navtrain` subset of NAVSIM, which provides **85k** real–raster paired samples. For training, we vary the fraction of real data kept ($\{1\%, 5\%, 20\%, 50\%, 100\%\}$) while replacing the remainder with rasterized samples, keeping the total size fixed. All models are evaluated on the validation split (**12k** real samples) using *MinADE*. Fig. 5 compares three settings: no alignment, spatial alignment, and spatial+global alignment. Both alignment strategies improve performance over the baseline, and combining spatial and global alignment yields the strongest results across all replacement ratios. This confirms that aligning synthetic and real features—both locally and globally—effectively reduces the domain gap and enables better transfer of supervision. Moreover, synthetic augmentation itself is beneficial: with 50% synthetic data, performance even surpasses training on 100% real samples. Overall, (i) R2R alignment systematically improves domain transfer, and (ii) large-scale rasterized data serve not only as a substitute but also as a powerful augmentation to limited real-world data.

**Ablation on Cross-Agent View Synthesis.** We next analyze the effect of scaling cross-agent synthesis, where trajectories of non-ego vehicles are rasterized to produce additional training samples. Starting from the `navtrain` split with **85k** real samples, we progressively add $\{1k, 10k, 100k, 500k, 1000k\}$ synthetic samples of other vehicles while keeping all other factors fixed. Figure 6 exhibits a clear log-scaling trend: the relation between sample count $x$ and MinADE $y$ is well fitted by $y = -0.021\ln(x) + 1.2173, \quad R^2 = 0.9942$, demonstrating diminishing but consistent gains as more synthetic samples are added. This log-scaling behavior closely mirrors findings from prior studies on data scaling in end-to-end driving (Baniodeh et al., 2025; Zheng et al., 2024). Notably, this finding is significant because the added data consists of *rasterized samples derived from other agents' trajectories*, showing that even secondary viewpoints contribute to scaling laws and improve planner robustness.

## 5 CONCLUSION

We presented **RAP**, a scalable framework for end-to-end driving that leverages lightweight 3D rasterization and feature-space alignment as an alternative to costly photorealistic rendering. By enabling recovery-oriented perturbations and cross-agent synthesis, RAP scales training to large-scale counterfactual scenes while preserving semantic and geometric fidelity. Extensive experiments across four benchmarks demonstrate that RAP consistently improves closed-loop robustness

and long-tail generalization, establishing a practical and effective recipe for scaling end-to-end autonomous driving. **Limitations & Future Work:** a key limitation of our approach is that it remains within the imitation learning paradigm, which inherits issues such as causal confusion. In future work, we aim to extend 3D rasterization into a full simulator to support closed-loop reinforcement learning, enabling richer interaction and policy improvement beyond offline demonstrations.

## ACKNOWLEDGMENTS

We would like to thank Ellington Kirby and Alexandre Boulch for their valuable help. The project was partially funded by Valeo, SwissAI, Sportradar, the Swedish Research Council (Vetenskapsrådet) under award 2023-00493, and the NAISS under award 2025/22-1173.

## ETHICS STATEMENT

This work adheres to the ICLR Code of Ethics. Our research focuses on developing scalable training methods for end-to-end autonomous driving planners using synthetic rasterized data. No human subjects were involved, and all datasets used (nuPlan, Waymo, NAVSIM, and Bench2Drive) are publicly released under their respective licenses. We acknowledge that autonomous driving research raises safety and fairness concerns. By emphasizing reproducibility, transparent methodology, and evaluation on diverse benchmarks, we aim to mitigate risks of overclaiming model capabilities. The proposed method does not introduce new privacy, security, or discrimination risks beyond those already present in the underlying datasets.

## REPRODUCIBILITY STATEMENT

We have made efforts to ensure that our work is reproducible. The paper provides detailed descriptions of the model architecture (Section 3), training setup (Section 4), and ablation studies (Section 4.2). Additional hyperparameters and implementation details are provided in the Appendix. All datasets used are publicly available (nuPlan, WOD, Bench2Drive), and we describe the curation and filtering process in Section 4. Source code and configuration files will be submitted and released with the camera-ready version to facilitate full reproducibility. These instructions apply to everyone, regardless of the formatter being used.

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

# A APPENDIX

## A.1 DISCUSSIONS

**Does Simplified 3D Rasterization Risk Overlook Critical Real-World Visual Cues?** Our 3D rasterization pipeline simplifies appearance by rendering only the annotated geometric elements of the scene, such as lanes, vehicles, and traffic signals. This choice reflects a deliberate trade-off. Rasterization gives us the speed and controllability needed to generate large-scale, diverse training data. High-fidelity rendering methods cannot reach this scale. At the same time, rasterization does not aim to replace real images. It complements them.

Despite this simplification, we find that the model retains strong perception and planning ability when applied to real-world images. Qualitative results on nuPlan (Figure 7) show that RAP correctly reacts to visual elements that are not present in the rasterization ontology, such as unannotated "Keep Left" signs or LED displays on a truck. Additional long-tail cases from WOD-E2E (Figure 8) further confirm that the model can handle subtle and unexpected signals at inference time.

These results indicate that simplified 3D rasterization does not hinder the model's ability to reason about fine-grained or OOD cues. Instead, it provides a strong geometric foundation that supports robust, scalable learning. Real images, seen both during training and always at inference, supply the remaining richness that rasterization cannot encode. The strong performance of RAP on the corner-case-focused WOD-E2E benchmark (Table 3) reinforces this conclusion.

**Does Real-to-Sim Feature Alignment Cause Information Loss?** A natural concern is that aligning real-world features to simplified rasterized features may suppress unannotated or subtle visual cues. Our results show that this does not occur, largely due to the **multi-task learning** design of our planner. The Real-to-Raster alignment provides a stable geometric prior, while the planning and perception objectives applied on real features compel the model to retain all task-relevant information necessary for accurate decision-making in complex traffic scenes.

To further understand this effect, we compare three alignment strategies: Raster-to-Real, symmetric alignment, and our Real-to-Raster variant. Following the same ablation setup in Section 4.2, we fix the real-data ratio to 50%. The results in Table 7 show that Real-to-Raster yields the lowest ADE, outperforming both alternatives. This supports our hypothesis that raster features serve as a clean structural scaffold, while the planning loss ensures that semantic richness from the real image is preserved. These findings align with recent representation alignment literature (Yu et al., 2024), where aligning to a well-structured feature space improves semantic abstraction without discarding perceptual detail. Future work may explore adaptive or task-aware alignment schemes to further reduce potential information loss.

Table 7: Comparison of alignment directions under 50% real data. Lower ADE is better.

| Alignment Variant | MinADE↓ |
|---|---|
| Raster-to-Real | 1.12 |
| Symmetric Alignment | 1.14 |
| Real-to-Raster (ours) | **1.02** |

**WOD Vision-based E2E Driving Finetuning.** For finetuning on the WOD-E2E Driving benchmark, we use a learning rate of $1 \times 10^{-5}$ and unfreeze the visual encoder. Finetuning is performed in two stages: first on the official training split, and then on the validation split with a batch size of 16. For the final test submission, we apply non-maximum suppression (NMS) ensembling over two checkpoints to improve robustness.

**Bench2Drive Mixed-Training and Dataset Alignment.** To enable mixed training with nuPlan and Bench2Drive, we align their data formats before batching. Specifically, we reorder camera views to a unified sequence, resize images to a common $576 \times 1024$ resolution, and adjust camera calibration matrices with rotation and scaling to match nuPlan's convention. Ego kinematics and target trajectories are normalized to the same coordinate frame. This alignment ensures that samples from both datasets share consistent geometry and calibration, allowing stable joint optimization.

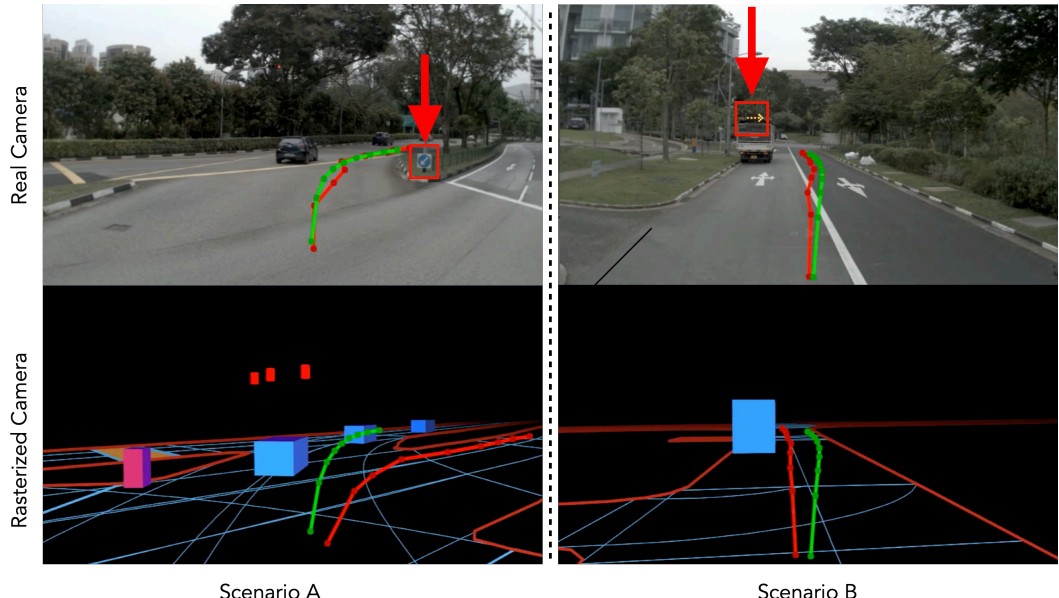

Figure 7: **RAP-DINO retains fidelity to fine-grained real-world cues.** This figure ablates our *single*, fully-trained model by comparing its predictions (red trajectory) against the ground truth (green trajectory) when conditioned on two different inputs: the raw real-world image (Top) vs. the simplified rasterized image (Bottom). **(Left) Scenario A:** An unannotated "Keep Left" sign, absent from the raster data, is correctly perceived from the raw image (Top), leading to a correct trajectory. The raster-conditioned model (Bottom) fails. **(Right) Scenario B:** A challenging OOD dynamic LED arrow, also absent from the raster, is successfully identified from the raw image (Top), resulting in a safe lane change.

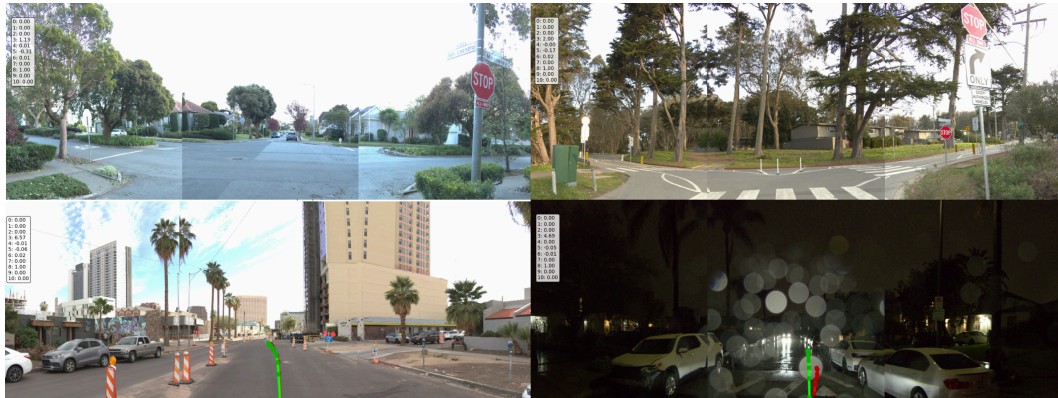

Figure 8: **Qualitative Results on the WOD Vision-Based E2E Driving Dataset.** The red trajectory shows the predictions of RAP-DINO, and the green trajectory denotes the ground-truth future path. Across the four examples, RAP-DINO successfully identifies a stop sign, traffic cones, and oncoming vehicles at night, leading to accurate and safe trajectory plans.

## A.2 VISUALIZATIONS FOR ABLATION STUDY ON 3D RASTERIZATION

To better illustrate the impact of different design choices in our 3D rasterization pipeline, we provide qualitative visualizations in Fig. 9. We compare object face rendering (colored vs. transparent), depth decay (enabled vs. disabled), and background style (pure black vs. natural sky–ground split). These examples highlight how each choice affects visual cues such as object semantics, depth perception, and training stability.

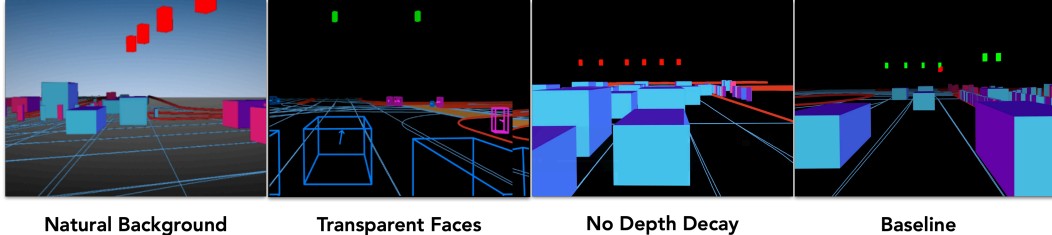

| Natural Background | Transparent Faces | No Depth Decay | Baseline |

Figure 9: **Qualitative comparison of 3D rasterization choices.** Columns correspond to different design choices: background (natural vs. black), face rendering (transparent vs. colored), and depth decay (off vs. on). The configuration with *colored faces + depth decay + black background (rightmost)* provides the most informative yet stable representation.

## A.3 LLM USAGE

In compliance with ICLR 2026 policy, we acknowledge the use of large language models (LLMs) in preparing this paper. LLMs were used to polish the writing, assist in retrieving related work, and provide limited support for brainstorming experimental designs. All research ideas, experiments, analyses, and conclusions are the sole work and responsibility of the authors.

Table 8: Training hyperparameters of RAP.

| Parameter | Value |
|---|---|
| Number of GPUs | $4 \times$ H100 |
| Total batch size | 128 for Navsim / 64 for WOD&Bench2Drive |
| Optimizer | AdamW (Loshchilov & Hutter, 2017) |
| Learning rate (initial) | 1e-4 |
| Learning rate schedule | Cosine Learning Rate Decay |
| Weight decay | 1e-4 |
| Epochs (pretraining) | 20 |
| Epochs (finetuning) | 20 |
| Gradient clipping | 0.0 |
| Domain alignment loss weights ($\lambda_{spatial}, \lambda_{global}$) | (0.002, 0.1) |
| Dropout | 0.1 |

## A.4 MORE TRAINING DETAILS

**Hyperparameters.** We summarize the training hyperparameters of RAP in Table 8. Unless otherwise noted, the same configuration is used across benchmarks, with dataset-specific finetuning schedules described in Sec. 4.

**Gradient Reversal Layer.** For adversarial global alignment, we adopt a gradient reversal layer following Ganin & Lempitsky (2015). During the forward pass, the GRL acts as an identity mapping, while in the backward pass it multiplies the gradients by $-\lambda$, forcing the encoder to learn domain-invariant features. We schedule $\lambda$ with a smooth annealing function

$$\lambda(p) = 0.1 * (\frac{2}{1 + \exp(-\gamma p)} - 1), \quad p \in [0, 1],$$

where $p$ is the training progress and $\gamma = 10$ by default. Our domain classifier is a lightweight MLP applied to aggregated features.

