# OpenReview forum: "RAP: 3D Rasterization Augmented End-to-End Planning"
_ICLR.cc/2026/Conference — ICLR 2026 Poster_

### Official Review · Reviewer_y7Na · 2025-10-16

**Soundness:** 4
**Presentation:** 4
**Contribution:** 3
**Rating:** 6
**Confidence:** 4

**Summary:**

This paper addresses the critical problem of brittleness in end-to-end autonomous driving models trained via imitation learning. The authors argue that this brittleness stems from the lack of "recovery" data, as models are only exposed to expert demonstrations. To overcome this, they propose Rasterization Augmented Planning (RAP), a scalable data augmentation framework.

The core contribution is a lightweight 3D rasterization pipeline that generates synthetic camera views from annotated geometric primitives (e.g., agent cuboids, lane polylines). This approach deliberately forgoes photorealism, arguing that semantic and geometric fidelity are sufficient for training robust driving planners, and that this method is far more scalable than computationally expensive alternatives like neural rendering or game engines.

This rasterization pipeline enables two key data augmentations: (1) **Recovery-oriented perturbations**, which simulate off-expert-path maneuvers to teach the policy how to recover from mistakes, and (2) **Cross-agent view synthesis**, which re-renders scenes from the perspective of other agents to dramatically increase the volume and diversity of training data.

To bridge the domain gap between the abstract rasterized views and real-world images, the paper introduces a Raster-to-Real (R2R) alignment module. R2R operates in the feature space, using a combination of spatial-level feature distillation (aligning real features to "cleaner" raster features) and global-level adversarial adaptation to enforce domain invariance.

The authors demonstrate the effectiveness of RAP through extensive experiments, achieving state-of-the-art results on four major benchmarks: NAVSIM v1/v2, the Waymo Open Dataset (WOD) Vision-based E2E Challenge, and Bench2Drive. The results show consistent improvements in closed-loop robustness and long-tail generalization, not only for their proposed model but also when RAP is applied to existing state-of-the-art planners.

**Strengths:**

1. **Pragmatic and Well-Motivated Core Idea:** The central thesis—that photorealism is not necessary for training robust E2E planners and that semantic/geometric fidelity is sufficient—is compelling and challenges a prevailing trend in the community. This provides a practical, scalable, and computationally efficient alternative to the prohibitively expensive photorealistic simulation pipelines (NeRFs, 3DGS, etc.). This insight is a significant conceptual contribution.
2. **Exceptional Experimental Validation:** The empirical evidence supporting the paper's claims is extensive and highly convincing.
    - **Breadth:** Achieving #1 ranking on four diverse and challenging benchmarks (NAVSIM, WOD, Bench2Drive) is a remarkable feat and strongly validates the method's effectiveness and generalizability.
    - **Rigor:** The evaluation covers both open-loop metrics (WOD) and, more importantly, pseudo-closed-loop (NAVSIM v2) and full closed-loop (Bench2Drive) simulations, directly addressing the core problem of closed-loop brittleness.
    - **Model-Agnosticism:** The authors demonstrate that RAP is not just a component of their specific model but a general framework that provides significant gains when applied to other SOTA models (e.g., `RAP-iPad`, `RAP-DiffusionDrive`). This greatly strengthens the paper's impact.
3. **Thorough and Insightful Ablation Studies:** The paper includes a comprehensive set of ablations that methodically validate the key design choices.
    - The study on rasterization design (Table 5) clarifies why specific choices like solid faces and depth decay are important.
    - The ablation on recovery perturbations (Table 6) directly links this augmentation to improved performance on the closed-loop-oriented NAVSIM v2 benchmark.
    - The analysis of the R2R alignment module (Fig. 5) clearly shows the benefit of both spatial and global alignment.
    - The scaling curve for cross-agent synthesis (Fig. 6) is particularly strong, demonstrating a clear log-scaling law that mirrors findings on real data scaling. This provides powerful evidence for the value of the generated synthetic data.
4. **High-Quality Presentation:** The paper is well-written, clearly structured, and easy to follow. The figures, especially the comparative illustration in Figure 1 and the system overview in Figure 2, are excellent and effectively communicate the core concepts.

**Weaknesses:**

1. **Oversimplification of the Static World:** The rasterization pipeline represents the world as a set of key primitives against a simple (often black) background. While the ablations show this is effective, it discards a vast amount of visual context from the static environment (buildings, foliage, non-annotated road signs, etc.). This simplification might be a vulnerability in complex urban scenes where such context is semantically important for driving decisions (e.g., navigating a complex, non-standard construction zone).
2. **Information Loss.** A primary weakness of the RAP framework is the inherent information loss resulting from its reliance on abstract 3D rasterization. By training the real-world encoder to align its features with a "clean" representation derived solely from annotated primitives (e.g., agent cuboids and lane lines), the model is implicitly taught to ignore any visual information not present in the labels. This poses a significant safety risk, as the system may fail to perceive critical, unannotated hazards such as road debris, potholes, temporary construction signs, or police officers directing traffic. Furthermore, this abstraction discards subtle but predictive visual cues, like wet road surfaces or holes in the road, which are crucial for nuanced, proactive driving. While this approach demonstrably improves robustness against common visual variations like weather and lighting, it does so at the cost of creating a potential "blindness" to novel or out-of-annotation long-tail events, a critical limitation for real-world deployment.
3. **RL Training.** A weakness is the paper's narrow focus on imitation learning, which fails to exploit the full potential of the proposed RAP framework. The authors have effectively built a fast and scalable driving simulator, an ideal environment for reinforcement learning (RL). By restricting their method to data augmentation for IL, they miss the opportunity to use RL to train an agent that could learn through active exploration, optimize for more complex rewards, and potentially discover policies superior to the original expert. This confinement to IL undersells the power of their core contribution, leaving its most transformative application as an RL training engine unexplored.

**Questions:**

1. Regarding the R2R alignment module: The spatial alignment loss (Eq. 2) updates the real features `F^r` to match the frozen raster features `F^s`, treating the latter as a "clean" supervision signal. Could you elaborate on the intuition behind this directional alignment? Did you experiment with a symmetric loss or alternative objectives where both feature distributions are jointly optimized?
2. Regarding the scaling laws (Fig. 6): The log-scaling behavior with cross-agent data is very promising. What do you theorize is the limiting factor for this scaling? Does the performance plateau come from the finite variety of agent behaviors in the log, or from the inherent domain gap that cannot be fully closed by R2R alignment?

---

> ### Author Response · Authors · 2025-11-18
>
> ## **W1: Oversimplification of the Static World**
>
> Thank you for raising this concern. We agree that rasterized data contains fewer semantics and a limited ontology, and this simplification is *intentional* as explained in Appendix A.1.
>
> Rasterized data is designed to **scale efficiently** and capture the **rules of the road**, structural geometry, and typical agent behaviors rather than full visual detail. This makes it well suited for generating large, diverse, and perturbation-focused trajectories.
>
> Importantly, rasterized scenes are not meant to **replace** real images. They **complement** them during training. The 3D rasterization offers a strong geometric prior and broad scenario coverage, while real images, used during training and always at inference, supply the fine-grained cues that rasterization cannot encode.
>
> As shown in the new qualitative results (Appendix A.1, PDF p.14), RAP-DINO correctly interprets cues that are **absent in the raster input**, such as:
> - Unannotated Keep Left signs
> - OOD LED hazard arrows
> - Night-time stop signs
> - Traffic cones and distant vehicles
>
> These examples confirm that although rasterized data omits such details, the model learns to recover them from real images at inference, illustrating the complementary roles of raster and real-image inputs.
>
> Overall, simplified rasterization is a **purposeful design choice** that enables scalable training while preserving the fine-grained perception needed for robust real-world closed-loop planning.
>
> ## **W2 & Q1: Information Loss and Intuition behind R2R alignment module**
>
> A natural concern is that aligning real features to simplified rasterized features might suppress subtle or unannotated information. Our results show this does **not** occur due to the **multi-task learning** structure of our planner:
>
> - The **Real to Raster alignment** offers a stable geometric prior.
> - The **planning and perception losses on real images** ensure the encoder preserves all task-relevant semantic information needed in complex scenes.
>
> In practice, raster features serve as a **clean structural scaffold**, and real-image supervision maintains full semantic richness.
>
> To directly evaluate whether alignment direction affects information retention, we compare three strategies under the same ablation setup:
>
> - Raster to Real
> - Symmetric alignment
> - Real to Raster (ours)
>
> With the real-data ratio fixed at 50 percent, results in Appendix A.1 show:
>
> | **Alignment Variant** | **MinADE↓** |
> |-----------------------|-------------|
> | Raster to Real        | 1.12        |
> | Symmetric             | 1.14        |
> | **Real to Raster (ours)** | **1.02** |
>
> Real2Raster achieves the best performance, supporting the intuition that rasterized features provide a well-structured prior that stabilizes learning, while the planning objective prevents loss of essential perceptual details.
>
> Both the qualitative evidence (fig. 7 and 8) and ablation studies confirm that Real to Raster alignment **does not cause harmful information loss**, and is the most effective and robust strategy among the tested variants.
>
> ## **W3: Narrow Focus on IL**
>
> We thank the reviewer for noting that RAP is trained within the imitation learning framework. Below we explain why IL was a deliberate starting point and how our method already demonstrates closed-loop robustness.
>
> IL enables us to **isolate and rigorously assess the 3D rasterization state representation** without the added complexity of full reinforcement learning. A full RL system would require:
>
> 1. **High-throughput GPU simulation** to sustain RL-scale interaction budgets.
> 2. **Stabilizing long-horizon closed-loop training**, including curriculum design and extensive tuning.
> 3. **Careful reward design**, which is difficult and highly sensitive.
>
> Even without full RL, our method shows **clear improvements in closed-loop robustness**, indicating that the 3D rasterization representation handles distribution shift effectively. As stated in the Conclusion, developing a full closed-loop RL system on top of RAP is a promising next step.
>
>
> ## **Q2. Regarding the scaling laws (Fig. 6): What do you theorize is the limiting factor for this scaling?**
>
> We believe there are two potential limiting factors for this scaling:
> 1. **Data Distribution Limitation**
>    As the dataset grows, it becomes dominated by common, low-diversity **routine trajectories**, which offer limited incremental learning value. Prior work on scaling laws for IL-based driving shows that **diversity**, rather than sheer volume, is key for continued gains. We believe a similar bottleneck applies here.
>
> 2. **Residual Domain Gap**
>    While our Raster-to-Real alignment reduces the gap, a small mismatch remains. The performance drop in Figure 5 when real-data proportion becomes too low (for example, 20 percent to 1 percent) indicates that a **minimal amount of high-quality real images** is still needed to anchor the feature space as the dataset scales.

---

### Official Review · Reviewer_HRhi · 2025-10-31

**Soundness:** 4
**Presentation:** 3
**Contribution:** 4
**Rating:** 8
**Confidence:** 4

**Summary:**

This paper propose that 3D rasterization of annotated primitives are semantically similar to images, so it can be used as input for end-to-end driving. By doing so, multiple data augmentation tricks can be applied to end-to-end driving, without costly image rendering. A raster-to-real alignment is further proposed to align the two representations in feature spaces.

**Strengths:**

* For the input of end-to-end driving models, a 3D rasterization representation is proposed to be the substitute of raw images , which is interestingly to already have similar semantics in DINOv3 feature space.
* With this representation, two types of augmentation are applied in the training of end-to-end driving model, with the need for costly image-space rendering in previous methods.
* A raster-to-real alignment module is proposed to enforce feature consistency between rasterized and real inputs at both spatial and global levels.

**Weaknesses:**

* On NAVSIM and WOD benchmarks, the method achieves SOTA performance. However, on the real close-loop benchmark Bench2Drive, the performance gain is incremental compared with baseline, which is contrary to Recovery-oriented perturbations.

**Questions:**

* Any rules considered in Recovery-oriented perturbations, for example, not collide with nearby vehicles?
* Why use minADE as metric in ablation study?
* In Fig.5, why the performance of 100% real data is not the best?

---

> ### Author Response · Authors · 2025-11-18
>
> We thank the reviewer for the thoughtful and insightful feedback. We address each weakness and question below.
>
>
>
> ## **W1: On Bench2Drive, the performance gain is incremental compared with baseline.**
>
> We acknowledge this observation regarding the incremental gain on the Bench2Drive closed-loop evaluation. We want to clarify that this outcome stems from the **cross-domain experimental setup**, rather than a limitation of the RAP representation itself.
>
> The Bench2Drive experiment uses a preliminary configuration where real-world NuPlan data (with perturbations) are simply mixed into the CARLA training set. This introduces two key bottlenecks:
> 1. **Missing Feature Alignment:** Bench2Drive data lack the paired 3D rasterized views required for our **Raster-to-Real Alignment Module**, preventing effective feature consistency.
> 2. **Large Domain Gap:** The gap between real-world NuPlan data and synthetic CARLA is substantial; without domain adaptation, naive mixing cannot fully exploit the added data.
>
> In contrast, our NAVSIMv2 ablation (Sec. 4.2) shows a clear **13.5% improvement** from recovery-oriented perturbation, demonstrating its effectiveness for strengthening closed-loop robustness. We will incorporate this discussion and acknowledge the incremental Bench2Drive gain in the paper’s limitation section to provide a more complete picture.
>
>
> ## **Q1. Any rules considered in Recovery-oriented perturbations, for example, not collide with nearby vehicles?**
>
> For the sake of simplicity and isolating the effect of the representation learning, we implemented **no post-filtering or rule-based constraints** (such as collision checks) in the creation of the Recovery-oriented Perturbations dataset. We only applied perturbations to the historical states of the ego-vehicle while keeping the future ground-truth trajectory unchanged.
>
> We agree with the reviewer that incorporating rule-based filtering (e.g., rejecting trajectories that result in immediate, obvious collisions) would further improve the data quality and potentially the resulting policy's performance.
>
> ## **Q2. Why use minADE as the metric in the ablation study?**
>
> Since our learning setup is strictly within the **Imitation Learning** framework, our ablation studies aim to measure how each architectural design (feature alignment, cross-agent views, etc.) improves the model’s ability to **imitate expert trajectories**. For this purpose, **minADE is a standard, intuitive, and widely used open-loop metric** that directly reflects how well the predicted trajectory matches expert demonstrations.
>
> In addition, prior work such as *Data Scaling Laws for End-to-End Autonomous Driving* also adopts **minADE as the primary indicator of data scaling trends**, reinforcing its suitability for isolating and analyzing the effects of each design choice within an IL setting.
>
> The only exception is the recovery-oriented perturbation experiment, where we report **EPDMS**, a **closed-loop evaluation metric** that measures how well the policy recovers and avoids compounding errors under perturbed states. Using EPDMS is essential for demonstrating that recovery-oriented perturbations truly improve **closed-loop robustness**, which cannot be captured by open-loop metrics like minADE. Moreover, we include Bench2Drive experiment in Sec 4.1, which is fully closed-loop.
>
> ## **Q3. In Fig. 5, why is the performance of 100% real data not the best?**
>
> This is a very insightful observation that highlights the strong **regularizing effect** of our 3D rasterization.
>
> Even though the mixed setup contains fewer real images, the abstract 3D raster scenes provide **clean, noise-free demonstrations** of the core driving factors (agent motion, interactions, collision risks, and simple road rules) without the distractions of textures, lighting variations, or background clutter. When trained jointly with real images, the model naturally splits the roles of the two modalities:
> - the rasterized scenes supply **stable geometric and motion priors**,
> - while the real images supply the **fine-grained visual cues** needed for perception.
>
> This combination reduces overfitting to incidental real-image patterns and pushes the model toward the true underlying driving dynamics. As a result, the mixed setup yields better generalization than using 100% real data alone.
>
> We will add a discussion of this interesting effect to the paper, as it is both unexpected and valuable for understanding the benefits of 3D rasterization.

---

### Official Review · Reviewer_eDtm · 2025-11-10

**Soundness:** 3
**Presentation:** 3
**Contribution:** 4
**Rating:** 6
**Confidence:** 3

**Summary:**

This paper presents an end-to-end driving framework that uses 3D rasterization instead of costly photorealistic rendering. By focusing on semantic and geometric fidelity and aligning synthetic with real data in feature space, RAP enables more flexible data augmentation and achieves robustness and generalization across multiple driving benchmarks.

**Strengths:**

- replaces expensive photorealistic rendering, offering a controllable way to generate diverse driving scenarios.
- provides experiments and ablations across multiple major benchmarks (NAVSIM, Waymo, Bench2Drive), consistently achieving strong results
- bridges synthetic and real data efficiently, reducing the sim-to-real gap without costly photorealistic rendering

**Weaknesses:**

- RAP remains within the imitation learning framework, so it still can display issues like causal confusion and lack of active policy improvement.
- the simplified rasterized scenes may miss fine-grained visual cues or rare real-world conditions that could matter for extreme edge-cases

**Questions:**

- do the authors plan to evaluate RAP outside of the IL setting? rl evaluations would demonstrate if the model can generalize under agent exploration
- can the authors provide additional examples showing reconstruction results on very OOD scenes? e.g. scenes with objects that have very little representation in the training dataset.

---

> ### Author Response · Authors · 2025-11-18
>
> We thank the reviewer for the thoughtful and insightful feedback. We address each weakness and question below.
>
>
> ## **Response to Weakness 1 and Question 1 (Limited to Imitation Learning)**
>
> We thank the reviewer for highlighting the limitation that RAP is currently trained within the imitation learning framework. We clarify below why IL was a deliberate choice and how our work already provides evidence toward closed-loop robustness.
>
> Our decision to begin with IL was intentional: it allowed us to **isolate and rigorously evaluate the effectiveness of the 3D rasterization state representation**, without conflating it with the numerous complexities introduced by full reinforcement learning. In particular, a full RL pipeline would require:
>
> 1. **High-throughput GPU-accelerated simulation** to sustain RL-scale interaction budgets.
> 2. **Stabilizing long-horizon closed-loop training**, including curriculum design and careful hyperparameter tuning.
> 3. **Designing a reward function suitable for real-world driving**, which is well-known to be non-trivial, often brittle, and highly sensitive to specification errors.
>
> Despite not performing full RL training in this paper, we emphasize that our method already demonstrates **meaningful gains in closed-loop robustness**. Specifically, our recovery-oriented perturbations serve as a controlled way of exposing the planner to off-nominal states, and the ablation results on NAVSIMv2 (Sec. 4.2) show a **substantial improvement in closed-loop metrics**, indicating that the 3D rasterization representation effectively handles distribution shift during online rollouts, even without RL.
>
> As stated in the Conclusion of our paper, developing a full closed-loop RL system on top of RAP is one of our most promising next steps and an active area of ongoing research.
>
>
> ## **Response to Weakness 2 and Question 2 (Simplified rasterized scenes may miss fine-grained visual cues.)**
>
> Thank you for highlighting this important concern. We agree that rasterized data, by design, lacks certain fine-grained semantics and has a more limited ontology. However, as clarified in **Appendix A.1**, this simplification is *intentional* and serves a specific role in our framework.
>
> First, rasterized data is designed to **scale efficiently** and capture the **rules of the road**, typical agent behaviors, and structural geometry, rather than full visual perception. This makes it extremely effective for producing large quantities of diverse and perturbation-oriented trajectories.
>
> Second, rasterized scenes are not meant to **replace** real images. Instead, they **complement** them during training. The 3D rasterization provides a strong geometric prior and exposes the model to a wide distribution of controllable, behavior-centric scenarios, while real images, seen both during training and always at inference, supply the remaining richness that rasterization cannot encode.
>
> As shown in newly added qualitative results (Appendix A.1, PDF p.14), RAP-DINO can correctly interpret cues that are **absent from the raster input**, including:
> - Unannotated *Keep Left* signs
> - OOD LED hazard arrows
> - Night-time stop signs
> - Traffic cones and distant vehicles
>
> These examples demonstrate that even though rasterized data omits such details, the model can recognize them at inference time from real images, confirming the complementary roles of rasterized and real-image inputs.
>
> Together, these findings support our claim that simplified rasterization is not a limitation but a **purposeful design choice** that enables scalable training while preserving the fine-grained capabilities required for robust, real-world closed-loop planning.

---

### Meta-Review · Area_Chair_3Dyt · 2026-01-02

**Summary:**

he submitted paper raises the question (and attempts to answer it ...) whether photorealistic rendering is necessary to learn driving policies from synthetic data. They work on a rasterized representation which discards details deemed unnecessary for driving, where Consistency in feature space between rasterized and "real" inputs is ensured with a dedicated module. This allows for dedicated data-augmentations in the simplified input space.

The idea is highly interesting and worthwhile exploring.
The performance of the method is high, with SoTA performance on NAVSIM and WOD.
The experimental validation was well appreciated with numerous studies.
The paper is well-written and presented.

Some minor weaknesses where raised: imitation learning only and no RL training; potential oversimplification (the AC judges that this weakness is sufficiently address by the experimental performance).

This paper was well received and there was consensus for acceptance.
The AC concurs.

**Reviewer Concerns:**

All (minor) concerns where addressed.

**Reviewer Scores:**

No discussion happened.

---

### Decision · Program_Chairs · 2026-01-26

Accept (Poster)